# Molecular Epidemiology and Antibiotic Resistance Associated with *Avian Pathogenic Escherichia coli* in Shanxi Province, China, from 2021 to 2023

**DOI:** 10.3390/microorganisms13030541

**Published:** 2025-02-27

**Authors:** Fangfang Li, Mengya Li, Lianhua Nie, Jiakun Zuo, Wenyan Fan, Liyan Lian, Jiangang Hu, Shuming Chen, Wei Jiang, Xiangan Han, Haidong Wang

**Affiliations:** 1College of Veterinary Medicine, Shanxi Agricultural University, Jinzhong 030801, China; kybgs408@163.com (F.L.); t20225019@stu.sxau.edu.cn (M.L.); z20223845@stu.sxau.edu.cn (L.N.); 15938272357@163.com (W.F.); 13834834183@163.com (S.C.); 2Shanghai Veterinary Research Institute, The Chinese Academy of Agricultural Sciences (CAAS), 518 Ziyue Road, Shanghai 200241, China; jkzuo1214@163.com (J.Z.); 18016565087@163.com (L.L.); vethjg@163.com (J.H.); jiangwei@shvri.ac.cn (W.J.)

**Keywords:** Shanxi province in China, avian colibacillosis, APEC, drug resistance, virulence gene, resistance gene

## Abstract

*Avian Pathogenic Escherichia coli* (APEC) constitutes a major etiological agent of avian colibacillosis, which significantly hinders the development of the poultry industry. Conducting molecular epidemiological studies of APEC plays a crucial role in its prevention and control. This study aims to elucidate the molecular epidemiological characteristics of *Avian Pathogenic Escherichia coli* in Shanxi Province. In this study, 135 APEC strains were isolated and identified from 150 liver samples of diseased and deceased chickens exhibiting clinical symptoms, which were collected from farms in Shanxi Province between 2021 and 2023. The isolates were then analyzed for phylogenetic clustering, drug resistance, resistance genes, virulence genes, and biofilm formation capabilities. The results revealed that the proportions of the A, B1, B2, and D evolutionary subgroups were 26.67%, 32.59%, 17.78%, and 15.56%, respectively. The drug resistance testing results indicated that 92% of the isolates exhibited resistance to cotrimoxazole, kanamycin, chloramphenicol, amoxicillin, tetracycline, and other antibiotics. In contrast, 95% of the strains were sensitive to ofloxacin, amikacin, and ceftazidime. The most prevalent resistance genes included tetracycline-related (*tetA*) at 88.15%, followed by beta-lactam-related (*bla-TEM*) at 85.19%, and peptide-related (*mcr1*) at 12.59%. The virulence gene analysis revealed that *ibeB*, *ompA*, *iucD*, and *mat* were present in more than 90% of the isolates. The results revealed that 110 strains were biofilm-positive, corresponding to a detection rate of 81.48%. No significant correlation was found between the drug resistance genes, virulence genes, and the drug resistance phenotype. A moderate negative correlation was observed between the adhesion-related gene *tsh* and biofilm formation ability (r = −0.38). This study provides valuable insights into the prevention and control of avian colibacillosis in Shanxi Province.

## 1. Introduction

*Avian Pathogenic Escherichia coli* (APEC) is responsible for both localized and systemic infections in poultry, presenting symptoms such as septicemia, subacute pericarditis, pneumocystitis, peripheral hepatitis, cellulitis, and oviductal infections [1,2]. APEC can infect poultry across a wide age range, but it is particularly prevalent in chicks, typically within the early stages of their life cycle. The incidence rate among chicks is notably high, ranging from 30% to 60%, and this infection often leads to a case fatality rate of 100% [3,4]. Consequently, the APEC-related poultry industry has incurred significant economic losses [5].

Recent studies have indicated that APEC is a potential foodborne zoonotic pathogen and serves as a reservoir for virulence and resistance genes of human Extraintestinal Pathogenic *Escherichia coli* (ExPEC) [6]. Specifically, APEC shares genetic similarities with ExPEC, Uropathogenic *Escherichia coli* (UPEC), and Neonatal Meningitis *Escherichia coli* (NMEC). APEC harbors the same virulence genes as UPEC and NMEC and can induce urinary tract infections and meningitis in mammals (e.g., mouse and rat models) [7,8]. Furthermore, the APEC-specific colicin V (ColV) plasmid has been detected in human ExPEC isolates, suggesting the potential for plasmid-mediated transfer of APEC’s virulence and resistance genes from poultry to humans [9].

The frequent and extensive use of antibiotics has led to an increase in the resistance of pathogenic bacteria and a decline in animal immunity, making animals more susceptible to bacterial diseases and resulting in significant economic losses [10]. Moreover, the irrational use of antibiotics has contributed to the emergence of numerous multidrug-resistant (MDR) bacteria, which pose a substantial threat to animal health and food safety [11], with resistance patterns that are complex and highly variable across regions [12]. APEC induces morbidity in poultry through various virulence factors, such as adhesins, invasins, protectins, iron acquisition systems, and serum resistance factors [13]. The virulence factors facilitate APEC adhesion, invasion, colonization, proliferation, immune evasion, and systemic dissemination, thereby enabling the establishment of APEC infection [14]. Additionally, APEC utilizes biofilm formation to resist antibiotics and sustain infection [15].

Biofilms are microcolony complexes of polysaccharides, proteins, nucleic acids, and lipids that enhance bacterial resistance and immune evasion. Studies have shown that bacteria encapsulated in biofilms are 1000 times more resistant than planktonic bacteria [16,17]. The formation of biofilms also complicates the treatment of APEC infections. Therefore, the detection and analysis of drug resistance, virulence genes, resistance genes, and biofilm formation abilities of APEC isolates in a specific region is crucial not only for preventing and controlling avian colibacillosis in poultry farms but also for mitigating the emergence of MDR.

In this study, we obtained 135 strains of *Avian Pathogenic Escherichia coli* by isolation and identification from 150 chickens suspected of avian colibacillosis in Shanxi Province, China. The phylogenetic grouping, drug resistance, virulence genes, drug resistance genes, biofilm-forming ability, and correlation among various factors were also analyzed, with a view to providing a clinical basis for the prevention and treatment of avian colibacillosis.

## 2. Materials and Methods

### 2.1. Bacterial Strains

Between January 2021 and June 2023, a total of 135 strains of *Avian Pathogenic Escherichia coli* (APEC) were isolated from 150 liver specimens sourced from sick or dead chickens on various poultry farms located in Shanxi Province, China.

### 2.2. APEC Isolation, Identification and DNA Extraction

Livers of ill or dead chickens were collected, and approximately 1 cm^3^ of liver tissue samples were cut in a sterile, ultra-clean room. Samples were enriched in buffered peptone water (Oxoid, Manchester, UK) and incubated at 37 °C for 24 h. The enriched bacterial cultures were streaked on eosin methylene blue (EMB) agar (Oxoid Company, Manchester, UK) using a sterile wire loop and incubated for 24 h at 37 °C. Colonies exhibiting a green metallic sheen were presumptively identified as *E. coli*. *E. coli MG1655* was used as a positive control strain.

Genomic DNA was extracted using the boiling method [14]. Briefly, 1.5 mL of suspended plaque samples were centrifuged for 15 min (13,523× *g*, 4 °C). The supernatant was discarded, and the pellet was resuspended in 1 mL of PBS. This washing step was repeated three times. The resuspended samples were incubated at 99 °C for 15 min. After centrifugation (15 min, 13,523× *g*, 4 °C), the genomic DNA supernatant was immediately cooled to −20 °C for storage.

The genomic DNA supernatant was used as a DNA template. DNA extracted from isolated *E. coli* strains was amplified using species-specific primers for phoA (F-CGATTCTGGAAATGGCAAAAG, R-CGTGATCAGCGGTGACTATGAC) and *E. coli*-specific primers for PCR [3]. DNA from *E. coli MG1655* served as the positive control, and PCR reactions without a DNA template served as the negative control. PCR products were analyzed by electrophoresis on 1% agarose gels, with gel images captured and analyzed using the GelDoc© Gel Documentation System (Bio-Rad, Hercules, CA, USA).

### 2.3. Identification of Phylogenic Groups

The phylogenetic groups of APEC isolates were determined using a triplex PCR targeting the genes *chuA*, *yiaA*, and *TspE4.C2*, as described by Clermont et al. [18]. A dichotomous decision tree was used to determine the phylogenetic group of an *E. coli* strain by using the results of the genes *chuA*, *yiaA*, and *TspE4.C2* (Appendix A). Genomic DNA from APEC isolates was used as the template. Primer sequences and related information are provided in Table 1. PCR reflecting conditions are as follows: denaturation for 4 min at 94 °C, 30 cycles of 5 s at 94 °C and 10 s at 59 °C, and a final extension step of 5 min at 72 °C.

### 2.4. Antimicrobial Susceptibility Testing

Antimicrobial susceptibility was tested using the Kirby–Bauer disk diffusion method on Mueller–Hinton agar (MHA) [19]. A total of 10 antibiotics were tested: chloramphenicol (CAP, 30 µg), florfenicol (FFC, 30 µg), amoxicillin (AMX, 20 µg), ceftazidime (CAZ, 30 µg), enrofloxacin (ENFX, 10 µg), ofloxacin (OFX, 5 µg), tetracycline (TC, 30 µg), cotrimoxazole (CTX, 19 µg), amikacin (AMK, 30 µg), and kanamycin (KAN, 30 µg). Antibiotic disks were obtained from Hangzhou Microbial Reagent Co., Hangzhou, China. *E. coli ATCC 25922* served as the quality control strain, with the interpretation of inhibition zone diameters adhering to the guidelines established by the Clinical and Laboratory Standards Institute (CLSI) [20]. Isolates exhibiting resistance to three or more distinct antibiotic classes were classified as multidrug-resistant (MDR).

### 2.5. Detection of Antimicrobial Resistance Genes and Virulence Genes

Followed established methods described in the relevant literature [21,22,23,24]. PCR was used to identify the presence of 11 antibiotic genes belonging to six different classes in all APEC isolates, with PCR reactions with an ultrapure water template serving as a negative control. These genes included β-lactams (*bla-TEM* and *bla-CTXM*), quinolones (*qnrA*), amphenicols (*cat1* and *floR*), sulfonamides (*sul1* and *sul2*), tetracyclines (*tetA*), aminoglycosides (*strA* and *aphA*), and polypeptide (*mcr 1*). Primer sequences and relevant information are provided in Appendix A.

PCR was also used to identify 17 virulence genes in APEC isolates, as described in the relevant literature [24,25]. These genes include adhesin-related genes (*aatA*, *papC*, *tsh*, *fimC*, *mat*, *vat*), iron transport-related factors *(fyua*, *iucD*, *irp2*), invasion-related genes (*ibeB, yijP*, *ibeA*), and serum resistance factors (*ompA*, *neuC*, *cva*, *iss*, *iroN*). Primer sequences for these genes are provided in Appendix A.

### 2.6. Determination of Biofilm Formation Ability

Reference utilized the crystal violet staining method to identify and analyze the biofilm of the APEC isolates [26]. The isolate was cultured in test tubes at 37 °C, with a concentration of OD_600_ = 1.0 in LB medium diluted to 1:100, and then added to a sterile 96-well cell culture plate. The culture was maintained in a stationary incubation at 37 °C for a duration of 24 h. Subsequent to incubation, the culture medium was discarded, followed by rinsing the wells twice with phosphate-buffered saline (PBS) and allowing them to dry. Subsequently, 200 microliters of 0.1% crystal violet solution was introduced, and the plate was further incubated at 37 °C for a period of 30 min. Following this, the crystal violet solution was discarded, and the wells were washed again twice with PBS and dried at room temperature. To dissolve the crystal violet, 200 µL of 95% ethanol was added, and the plate was incubated at 37 °C for 10 min. The absorbance value of OD_595_ was then measured using an enzyme marker, and the results were recorded and analyzed.

The criteria for biofilm formation were as follows [27]:

Strong biofilm formation: OD595 > 4 × ODc;Moderate biofilm formation: 2 × ODc ≤ OD595 ≤ 4 × ODc;Weak biofilm formation: ODc ≤ OD595 < 2 × ODc;No biofilm formation: OD595 ≤ ODc.

### 2.7. Statistical Analyses

SPSS statistics version 26.0 and GraphPad Prism version 8.0 software were used for statistical analyses. The Spearman correlation coefficient (r), serving as a non-parametric measure of the strength of association between two variables, ranges in value from −1 to 1. The assessment of correlation was conducted using the chi-square test and by computing Spearman’s rank correlation coefficient (r). Specifically, an absolute value of r within the range of 0.5 to <1.0 denoted a high level of correlation, 0.3 to <0.5 indicated a moderate correlation, 0.1 to <0.3 signified a low correlation, and an absolute value of r < 0.1 was indicative of no correlation, according to the previous literature [28].

## 3. Results

### 3.1. Phylogenetic Grouping

The distribution of *chuA*, *yiaA*, and *TspE4.C2* in APEC isolates was examined separately, and the phylogenetic subgroups of isolates were determined based on the gene distribution combinations. The results are shown in Figure 1.

### 3.2. Antibiotic Susceptibility

The antibiotic susceptibility testing showed that 135 APEC isolates of *E. coli* were 100% resistant to cotrimoxazole and kanamycin (Table 2). The resistance rates to chloramphenicol (98.52%, 133/135), tetracycline (98.52%, 133/135), and amoxicillin (92.59%, 125/135) were very high, respectively. Moreover, the resistance rates followed for florfenicol (69.62%, 94/135) and enrofloxacin (14.07%, 19/135). It is noteworthy that all 135 strains of *Avian Pathogenic Escherichia coli* (APEC) exhibited resistance to at least four distinct antibiotic classes, thereby classifying them as multidrug-resistant (MDR) strains. The results are shown in Figure 2.

### 3.3. Biofilm-Forming Ability

The 135 APEC isolates were identified using crystal violet staining, and the results showed that 81.48% (110/135) of the strains were positive for biofilm formation. Among these, 46.47% (63/135) exhibited strong film-forming ability, 23.70% (32/135) exhibited moderate film-forming ability, and 11.11% (15/135) exhibited weak film-forming ability.

### 3.4. Analysis of Drug Resistance Genes in APEC

The results of APEC drug resistance gene testing are shown in Figure 3. The *tetA* gene, related to tetracyclines, had the highest detection rate at 88.15%. The *cat1* and *flo* genes, related to aminoglycosides, had detection rates of 2.22% and 76.30%, respectively. The *bla-TEM* and *bla-CTX-M1* genes, related to β-lactams, had detection rates of 85.19% and 38.52%, respectively. The detection rates of *strA* and *aphA*, genes related to aminoglycosides, were 67.41% and 39.26%, respectively. The *sul1* and *sul2* genes, related to sulfonamides, had detection rates of 22.96% and 41.48%, respectively. Finally, the *mcr1* gene, related to peptides, was detected at a rate of 12.59% (Figure 3).

### 3.5. Analysis of Virulence Genes in APEC

A total of 17 virulence genes were detected in 135 APEC isolates, with significant differences in key virulence genes between strains. The virulence genes with detection rates higher than 90% included *mat* (93.33%, 126/135), *iucD* (91.11%, 123/135), *ompA* (97.78%, 132/135), and *ibeB* (99.26%, 134/135). The detection rate of the three iron transport-related factor genes *fyua*, *icuD*, and *irp2* exceeded 50%, and the serum resistance factor-related gene *ibeB* reached 99.26% in these 135 APEC strains. *PapC* was only present in a few strains, and the detection rates of *vat* and *ibeA* were lower, at 5.93% and 20.74%, respectively (Figure 4a). A comparison of the distribution of six virulence genes with detection rates exceeding 89% among *Escherichia coli* isolates exhibiting different biofilm-forming capabilities revealed the highest prevalence of *fimC*, *mat*, *iucD*, *ompA, ibeB*, and *yijP* in strains with strong biofilm-forming abilities, exceeding 50%. In strains lacking biofilm-forming ability, the detection rates of these six genes ranged from 20% to 30%. Strains with intermediate and weak biofilm-forming abilities exhibited similar detection rates for these six genes, approximately 10% to 20% (Figure 4b).

### 3.6. Correlations Between Antibiotic-Resistant Phenotypes and Biofilm-Forming Ability

The 135 APEC isolates exhibited a correlation between biofilm-forming ability and antibiotic resistance. The biofilm-forming ability of the strains was positively correlated with ofloxacin and amikacin (r = 0.123, r = 0.201) but negatively correlated with chloramphenicol, florfenicol, and amoxicillin (r = −0.103, r = −0.203, r = −0.1) (Figure 5).

### 3.7. Correlations Between Antibiotic-Resistant Phenotypes and Resistant Genes

The correlation between the resistant genes carried by 135 APEC isolates and the antibiotic resistance phenotypes was analyzed. The results are presented in Figure 6. Among the acylated alcohol antibiotics, florfenicol exhibited a low negative correlation with *cat1* (r = −0.13), a low correlation with the β-lactam gene *blaCTX-M1* (r = 0.16), and a low positive correlation with the sulfonamide resistance gene *sul2* (r = 0.19). The chloramphenicol resistance phenotype in acyl alcohol antibiotics exhibited no correlation with resistance genes but a low correlation with the mobile genetic progenitor-related gene *int* (r = 0.18). Among the β-lactam antibiotics, both amoxicillin and ceftazidime displayed a negative correlation with *bla-TEM* (r = −0.11, r = −0.04) but a positive correlation with *blaCTX-M1* (r = 0.12, r = 0.10). Ceftazidime exhibited a low positive correlation (r = 0.13, r = 0.17, r = 0.16, r = 0.15) with the amidohydrin gene *fioR*, the aminoglycoside gene *strA*, the polypeptide gene *mcr1*, and the mobile genetic element gene *int*. Amoxicillin did not show a significant correlation with other resistance genes.

Among quinolone antibiotics, the enrofloxacin and ofloxacin resistance phenotypes did not show a significant correlation with the quinolone resistance gene *qnrA*. Enrofloxacin exhibited a low positive correlation with the β-lactam gene *blaCTX-M1*, the aminoglycoside gene *aphA*, and the sulfonamide resistance genes *sul1* and *sul2* (r = 0.16, r = 0.13, r = 0.13 and r = 0.19). Ofloxacin exhibited a low positive correlation with the aminoglycoside gene *strA* (r = 0.24).

Among tetracyclines, the tetracycline resistance phenotype did not show a significant correlation with the tetracycline resistance gene *tetA*; however, tetracycline exhibited a low negative correlation with the aminoglycoside gene *aphA*, the sulfonamide resistance gene *sul2*, and the polypeptide gene *mcr1* (r = −0.15, r = −0.15, r = −0.14).

Among aminoglycoside antibiotics, resistance phenotypes correlated with resistance genes, with amikacin exhibiting a low positive correlation with the *strA* and *aphA* genes (r = 0.15, r = 0.19), a low positive correlation with the tetracycline-resistant gene *tetA* (r = 0.16), a low positive correlation with the amidohydrin gene *fioR* (r = 0.13), a low positive correlation with the sulfonamide-resistant genes *sul1* and *sul2* (r = 0.18, r = 0.20), and a low positive correlation with both the gene *mcr1* and the gene *int* (r = 0.23, r = 0.16).

### 3.8. Correlations Between Antibiotic-Resistant Phenotypes and Virulence Genes

The correlation between the virulence genes carried by 135 APEC isolates and their corresponding antibiotic resistance phenotypes was analyzed (Figure 7). The results revealed that, among the adhesion-related genes, the carriage of *aatA* exhibited a low positive correlation with the resistance rate to ofloxacin (r = 0.16, r = 0.13); the carriage of *tsh* exhibited a low positive correlation with the resistance rates of florfenicol and enrofloxacin (r = 0.25, r = 0.14); the carriage of *vat* demonstrated a low negative correlation with the resistance rate of enrofloxacin (r = −0.16), while its correlation with the amikacin resistance rate was positively low (r = 0.22). Among the invasion-related genes, the carriage of *yijp* exhibited a low positive correlation with the resistance rates to ofloxacin and amikacin (r = 0.17, r = 0.16), while the carriage of *ibeA* exhibited a low positive correlation with the resistance rate of florfenicol (r = 0.26), as well as a low negative correlation with the resistance rates of ceftazidime and ofloxacin (r = −0.11, r = −0.11). Among the antisera survival factor-related genes, the carriage of *neuC* exhibited a low positive correlation with the resistance rate to florfenicol (r = 0.29), and the carriage of *cva* exhibited a low positive correlation with the resistance rates of florfenicol and enrofloxacin (r = 0.29, r = 0.21). Among the iron transport-related factors, the carriage rate of *fyua* exhibited a low positive correlation with the resistance rate of amikacin (r = 0.13), and the carriage rate of *irp2* exhibited a low negative correlation with the resistance rate of ceftazidime (r = −0.14).

### 3.9. Correlations Between Biofilm Formation Ability and Virulence Genes

This study also analyzed the correlation between virulence genes carried by 135 APEC strains and their biofilm formation ability. As illustrated in Figure 8, numerous virulence factors exhibited a weak negative correlation with biofilm formation ability, including the adhesion-related genes *tsh*, *fimC*, and *mat* (r = −0.34, r = −0.11 and r = −0.11), the invasion-associated gene *ibeA* (r = −0.26), the serum resistance factor-related genes *neuC*, *cva*, and *iss* (r = −0.16, r = −0.28, and r = −0.15), as well as the iron transport-related factor *iroN* (r = −0.17). However, the prevalence of *aatA* among adhesion-related genes and *ompA* among serum resistance factor-related genes demonstrated a weak positive correlation with biofilm formation ability (r = 0.13, r = 0.13).

## 4. Discussion

APEC serves as a reservoir for virulence and resistance genes in human Extraintestinal Pathogenic *Escherichia coli* (ExPEC), exhibiting genomic structural similarities and shared evolutionary and ecological characteristics with human-derived *E. coli*. Najafi et al. demonstrated that highly pathogenic *E. coli* strains in ExPEC predominantly belong to the B2/D phylogenetic group [28]. A similar study by Johnson et al. involving 994 poultry isolates found that most ExPEC strains were associated with the *E. coli* phylogenetic groups B2/D [29]. Liu et al. tested 122 clinical isolates from dead and diseased chickens collected from broiler farms in Hunan Province and found that highly pathogenic strains primarily belonged to the B2 and B1 phylogenetic groups [30]. The proportions of the B2 and D phylogenetic subgroups in this study were 19.35% and 11.83%, respectively, indicating a potential zoonotic risk associated with these isolated strains.

Antibiotic resistance poses a significant threat to both human medicine and the poultry industry. The use of antibiotics in animal husbandry has been shown to elevate the risk of resistance in both commensal and pathogenic enteric bacteria in food animals [9]. APEC has developed resistance mechanisms against nearly all classes of antibiotics typically employed in poultry medicine, with notable exceptions including carbapenems, imipenem, streptomycin, and tetracycline [31,32,33]. The widespread use of antibiotics in poultry farming has been identified as a major contributing factor to the emergence and persistence of antibiotic-resistant APEC strains. The overuse and misuse of antibiotics, particularly in intensive farming systems, have created selective pressures that favor the survival and proliferation of resistant strains, which in turn complicates the treatment and control of infections. APEC strains collected from poultry farms in southern and central China, for example, have demonstrated significant resistance to commonly used antibiotics, such as kanamycin, tetracycline, chloramphenicol, and amoxicillin [34,35,36]. In fact, resistance rates to these antibiotics often exceed 90%, with some isolates showing resistance to multiple antibiotic classes simultaneously. This level of resistance is a direct reflection of the selective pressures exerted by the extensive and often inappropriate use of these antibiotics in poultry production. In this study, all 135 APEC strains from Shanxi, China, exhibited multidrug resistance between 2021 and 2023, with at least fourfold resistance, and one isolate was resistant to all 10 antibiotics. All strains exhibited high resistance to kanamycin, cotrimoxazole, tetracycline, chloramphenicol, and amoxicillin, with only ofloxacin, amikacin, and ceftazidime proving effective against more than 90% of the APEC strains.

*E. coli* exhibits a strong capacity for accumulating drug resistance genes [37] and transferable resistance genes, which contributes to its significant drug resistance [38]. Many drug-resistant genes can be transferred horizontally between different strains through mobile genetic elements, such as plasmids, rendering sensitive strains resistant [39]. The resistance gene detection results in this study revealed that the tetracycline-associated resistance gene *tetA* had the highest detection rate, while several other antibiotic-associated resistance genes exhibited higher detection rates, with the exception of the quinolone *qnrA* and aminoglycoside cat genes, which had lower detection rates. In contrast, the drug sensitivity test results indicated that the resistance rate to tetracycline was 98.92%. Conversely, the resistance rates for the quinolones enrofloxacin and ofloxacin were below 13%, and the resistance gene results were positively correlated with the phenotypic resistance outcomes.

Virulence gene analysis revealed that among the adhesin-related genes, the expression levels of the *fimC* and *mat* genes were the highest, approximately 90%. The *fimC* gene regulates the assembly of the bacterial epithelial membrane, and the FimC protein acts as a chaperone for the FimH protein, playing a crucial role in maintaining the folding and length of type I bacterial fimbriae while enhancing the immunogenicity of the FimH protein. The *fimC* gene is crucial for bacterial adhesion and immune response [40]. Deletion or mutation of this gene typically results in the blockage of type I bacterial fimbriae synthesis, thereby affecting bacterial adhesion and biofilm formation ability [41]. *Mat* gene expression not only affects bacterial adhesion but is also closely linked to its pathogenicity and transmissibility. During infections, such as urinary tract infections, bacteria harboring the *mat* gene are more likely to adhere to the surface of bladder epithelial cells, facilitating the onset and spread of infection. The expression of the *mat* gene was found to be highly correlated with the *fimC* gene, suggesting their interaction or synergy in participating in the pathogenic process of bacterial strains [42].

Among the iron transport-related genes, the *iucD* and *ompA* genes had the highest detection rates. These genes may contribute to biofilm formation and maintenance by promoting the uptake and transport of iron ions, thereby providing essential iron to bacteria within the biofilm [43]. The expression levels of these genes may also indirectly affect the drug resistance of bacteria within the biofilm by regulating the iron metabolism process [44]. Deletion of *ompA* has been shown to enhance the formation of the epidermal membrane, possibly due to a compensatory mechanism that occurs during strain growth, leading to the upregulation of other genes involved in membrane formation [45]. However, it has also been demonstrated that *ompA* deletion may reduce the bacteria’s ability to adhere to host cells, thus indirectly affecting periplasmic formation [46]. This is because epidermal formation typically begins with bacterial adhesion to the host cell, and OmpA plays a crucial role in this process.

The invasion-related gene *ibeB* encodes the IbeB protein, which plays a dual role in bacterial pathogenesis. On the one hand, IbeB facilitates bacterial invasion and colonization of host cells or the extracellular matrix while also enhancing bacterial resistance to antimicrobial agents. Conversely, it plays a pivotal role in biofilm formation and maintenance [47]. The protein encoded by the *yijP* gene is involved in the invasion of human brain microvascular endothelial cells by *Escherichia coli* and is a key factor in bacterial traversal of the blood–brain barrier, playing a significant role in pathogenicity [48]. During biofilm formation, the YijP protein may also facilitate bacterial adhesion and colonization of host cells or the extracellular matrix, processes that are fundamental to biofilm formation and critical for bacterial survival and proliferation in specific environments.

In the initial analysis, the correlation between antibiotic resistance phenotypes and resistance genes, as well as virulence genes, respectively, was investigated. The results demonstrated that both correlations exhibited low significance (|r| < 0.3). We further examined the correlation between virulence genes carried by 135 APEC strains and their biofilm formation ability. The results revealed an intriguing phenomenon: most virulence genes displayed a low negative correlation with biofilm formation capability (|r| < 0.3). However, a moderate negative correlation was observed between the adhesion-related gene *tsh* and biofilm formation ability (r = −0.38). The gene *tsh* encodes the temperature-sensitive hemagglutinin Tsh, which plays a pivotal role in the pathogenesis of APEC. Tsh not only possesses adhesive properties, facilitating bacterial colonization and reproduction at the site of infection, but may also be associated with bacterial iron uptake capabilities [49,50]. This finding may suggest that, under certain circumstances, APEC may sacrifice virulence traits under selective pressure to enhance its survival and dissemination abilities in the host environment by promoting biofilm formation. In subsequent research, we will delve deeper into the specific role of Tsh in the pathogenesis of APEC and its interaction with the host immune system, providing a theoretical basis for the development of novel vaccines and antimicrobials targeting APEC.

## 5. Conclusions

In this study, 135 strains of *Avian Pathogenic Escherichia coli* from Shanxi, China, were analyzed for phylogenetic grouping, drug resistance, epidermal membrane formation capability, drug resistance genes, and virulence gene detection. In terms of virulence, the isolates carried a variety of virulence genes, and there were more strains of the strong biofilm type, which made prevention and control very difficult. In terms of drug resistance, the isolates carried multiple drug resistance genes, which led to a severe drug resistance situation in the region, mainly sensitive to ceftazidime and amikacin, which can be reasonably used in treatment and prevention. In conclusion, this study provides theoretical data for the prevention and control of *Avian Pathogenic Escherichia coli* disease in some areas of Shanxi Province.

## Figures and Tables

**Figure 1 microorganisms-13-00541-f001:**
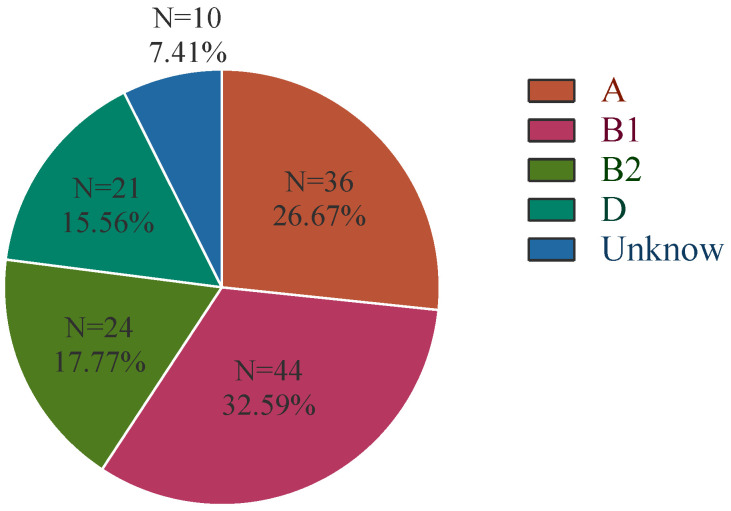
Phylogenetic clustering of APEC isolates.

**Figure 2 microorganisms-13-00541-f002:**
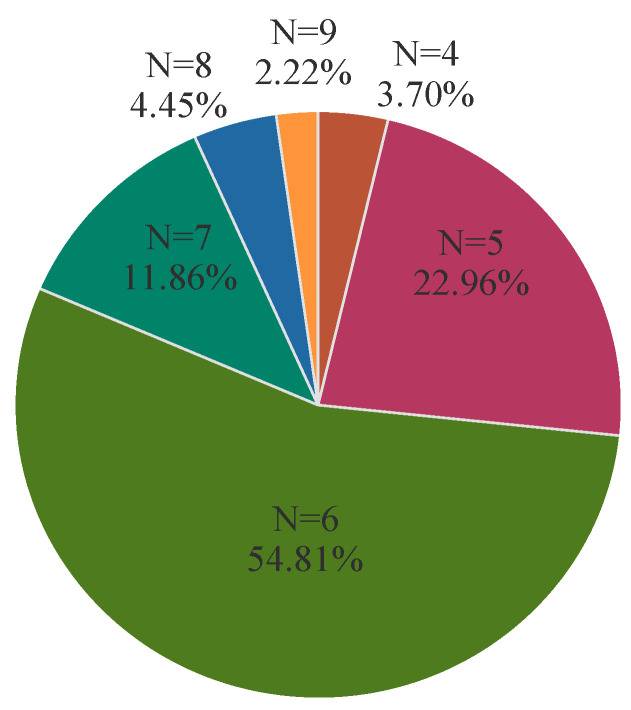
Multidrug resistance in APEC isolates. The color coding denotes the number of resistant weights of APEC isolates.

**Figure 3 microorganisms-13-00541-f003:**
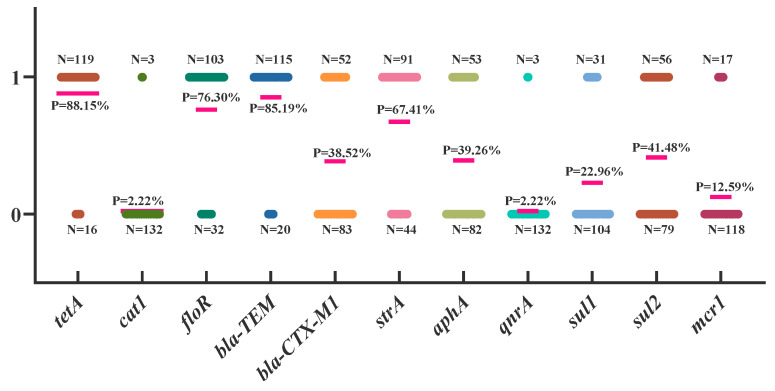
Detection of resistance genes. Each dot in the graph represents an APEC isolate. A value of 0 on the vertical axis indicates no detection, while 1 indicates detection. The horizontal line indicates the overall detection level of the gene.

**Figure 4 microorganisms-13-00541-f004:**
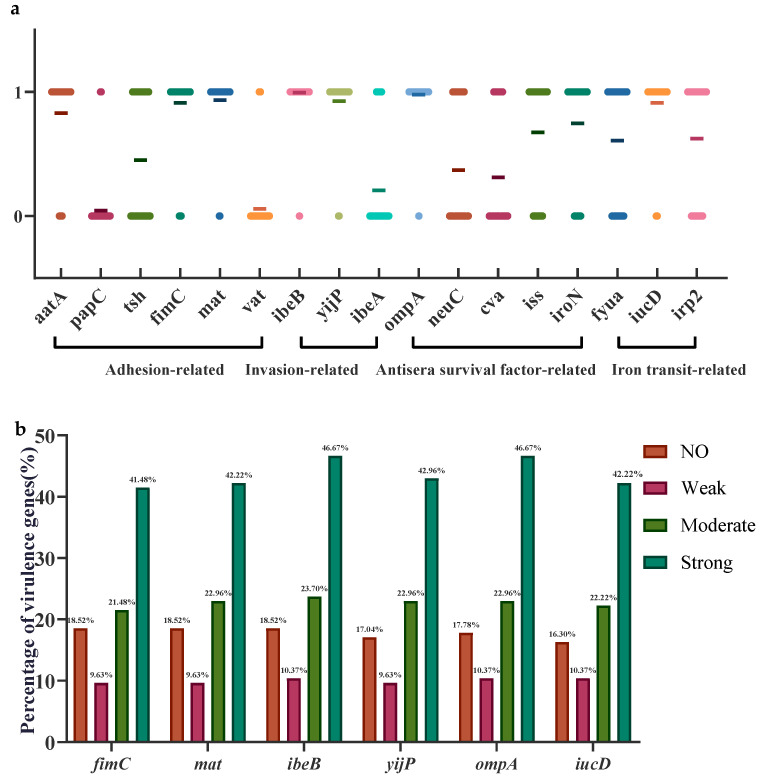
Analysis of virulence genes in APEC. (**a**) Detection of individual genes. Each point on the graph represents an APEC strain. A value of 0 on the vertical axis indicates no detection, while 1 indicates detection. The horizontal line denotes the overall detection rate of the corresponding gene. (**b**) Distribution of six virulence genes with detection rates exceeding 89% in APEC strains with different biofilm-forming abilities.

**Figure 5 microorganisms-13-00541-f005:**
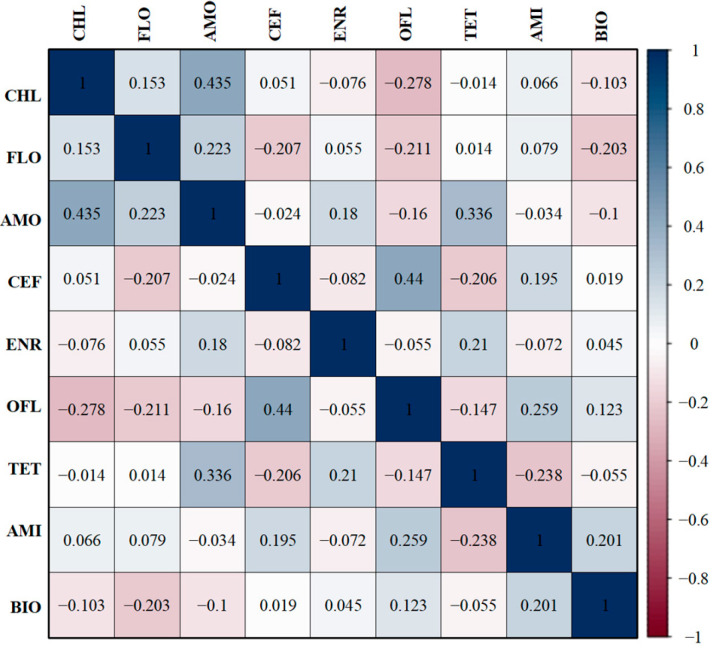
Correlation of antibiotic-resistant phenotypes and biofilm-forming ability. The intensity of the color represents the value of the correlation coefficient (r). CHL—chloramphenicol, FLO—florfenicol, AMO—amoxicillin, CEF—ceftazidime, ENR—enrofloxacin, OFL—ofloxacin, TET—tetracycline, AMI—amikacin, BIO—biofilm-forming ability.

**Figure 6 microorganisms-13-00541-f006:**
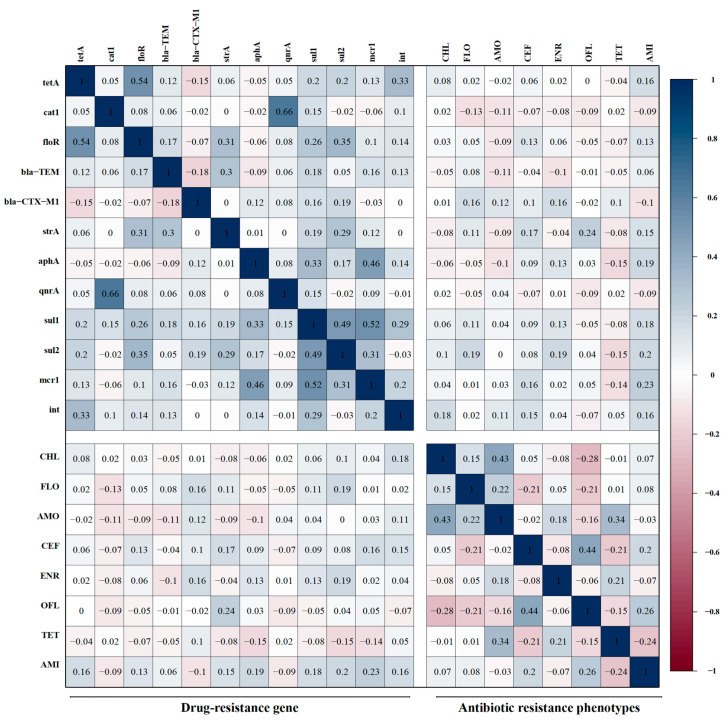
Correlation of drug-resistant genes and antibiotic resistance phenotypes. The intensity of the color represents the value of the correlation coefficient (r).

**Figure 7 microorganisms-13-00541-f007:**
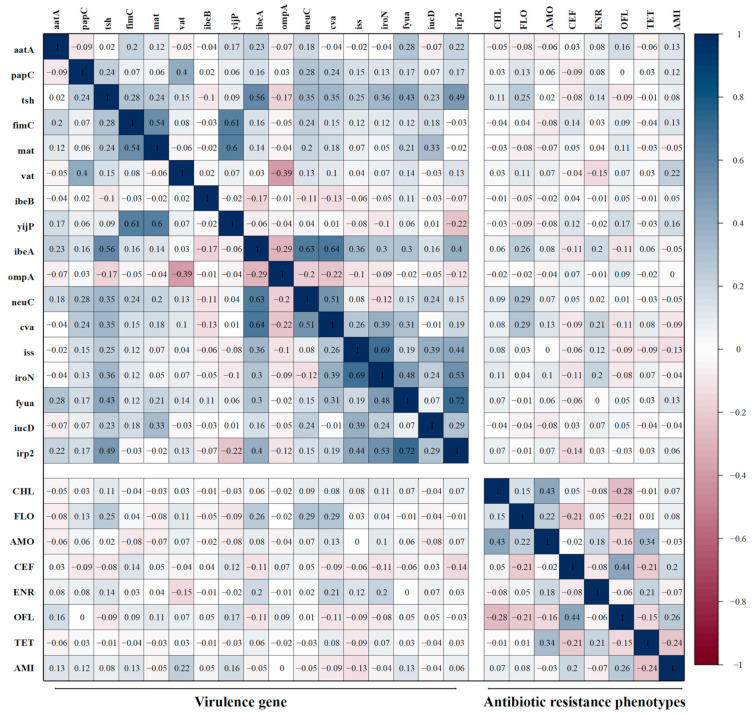
Correlation between virulence genes and antibiotic resistance phenotypes. The intensity of the color indicates the value of the correlation coefficient (r).

**Figure 8 microorganisms-13-00541-f008:**
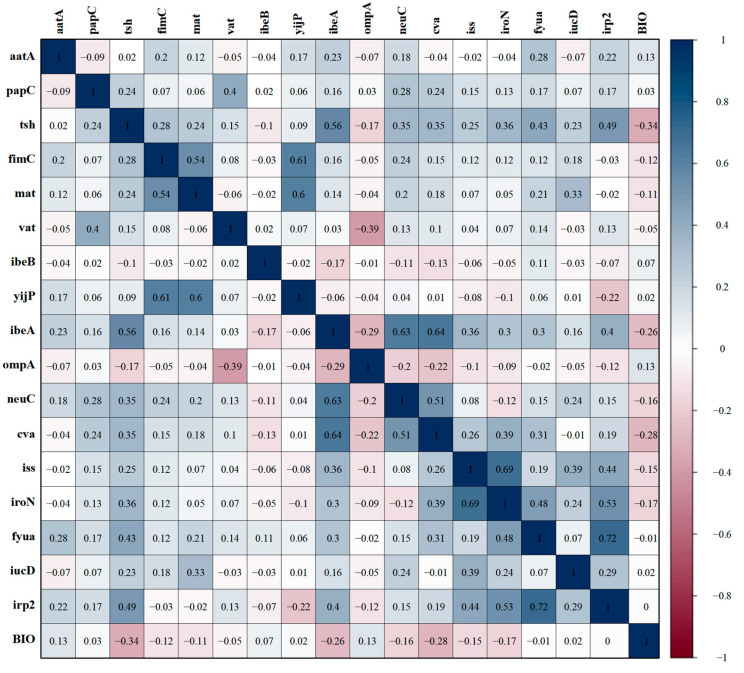
Correlation between virulence genes and biofilm formation ability. The intensity of the color indicates the value of the correlation coefficient (r).

**Table 1 microorganisms-13-00541-t001:** Primers for detection of phylogenetic clustering genes of APEC.

Gene	Primer	Sequence	Product Length (bp)
*yiaA*	yiaA-F	TGAAGTGTCAGGAGACGCTG	211
	yiaA-R	ATGGAGAATGCGTTCCTCAAC	
*chuA*	chuA-F	GACGAACCAACGGTCAGGAT	279
	chuA-R	TGCCGCCAGTACCAAAGACA	
*TspE4.C 2*	TspE4-F	GAGTAATGTCGGGGCATTCA	152
	TspE4-R	CGCGCCAACAAAGTATTACG	

**Table 2 microorganisms-13-00541-t002:** Antibiotic susceptibility of APEC isolates from chickens.

Antibiotics Class	Antibiotics Name	Number of Positive Isolates			Percentage of Resistance
		Sensitivity	Intermediate	Resistance	
Amphenicols	Chloramphenicol	1	1	133	98.52%
	Florfenicol	16	25	94	69.62%
Beta-lactams	Amoxicillin	2	8	125	92.59%
	Ceftazidime	111	20	4	2.96%
Quinolones	Enrofloxacin	24	92	19	14.07%
	Ofloxacin	96	33	6	4.44%
Tetracyclines	Tetracycline	2	0	133	98.52%
Sulfonamides	Cotrimoxazole	0	0	135	100%
Aminoglycosides	Amikacin	98	28	9	6.66%
	Kanamycin	0	0	135	100%

## Data Availability

The original contributions presented in this study are included in the article/Appendix A. Further inquiries can be directed to the corresponding authors.

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
