# Peer review of "Molecular Epidemiology and Antibiotic Resistance Associated with Avian Pathogenic Escherichia coli in Shanxi Province, China, from 2021 to 2023"

_microorganisms, 2025, doi:10.3390/microorganisms13030541_

Round 1

Reviewer 1 Report

Comments and Suggestions for Authors

There are editorial issues in the entire manuscript, please address them.

M&Ms section: should be improved.

Results section: some parts are difficult to follow, please rewrite

Discussion: Biofilm results were not sufficiently discussed. 

Comments on the Quality of English Language

English is ok but some parts are very difficult to follow.

Reviewer 2 Report

Comments and Suggestions for Authors

The authors in  this study provided theoretical data for the prevention and control of avian pathogenic E. coli disease in some areas of Shanxi Province.

It is a good representation a problematic issue regarding amr resistence in pultry industry.

I recommend to improve conclusion and explore the use od autogenous vaccine to fight amr resistance.

Please check all the italic for E. coli in the text.

Reviewer 3 Report

Comments and Suggestions for Authors

The manuscript “Molecular Epidemiology and Antibiotic Resistance Associated with Avian Pathogenic Escherichia coli of Shanxi Province in China From 2021 to 2023” by Fangfang Li and coauthors conducted an epidemiological study on APEC strains in chickens from China. Using PCR, they performed Clermont typing and detection of virulence and resistance genes. In addition, they performed phenotypic tests for resistance and biofilm formation capacity. In general, the study is not very new and has several flaws. The authors omit important methodological details that limit reproducibility. Furthermore, they carried out correlation analyzes between different parameters without an appropriate foundation, justification and subsequent discussion. In times with NGS availability, detecting only the presence or absence of genes is very limited, especially to offer correlation analysis.

Introduction

Add information on the genetic aspects that confer the ability of APEC to cause disease in chickens unlike the other types of E. coli.

Since the Clermont scheme is used in the study, I suggest providing information about it.

Include the objectives of this study at the end of the introduction.

Methods

Bacterial Strains, Broilers, and Diets: This section requires improvement. It mentions diets in the heading but there is no information about them in the text. The authors should give more details of the farms (number, ages,...), the clinical symptoms observed and the collection samples procedures. Furthermore, the logical order of procedures must be maintained. The number of APEC strains isolated should be reported in results.

Line 82: Uniformize the writings. E. coli should be written in italics whenever it is mentioned.

Lines 83-85: Cite the references of the primers and reactions used.

Section 2.3: It is redundant and breaks the logical order of the procedures. Should be merged with section 2.2.

Detection of antimicrobial resistance genes: The authors mention that they tested 10 genes by PCR but list 11 in the first paragraph. Check.

Sections 2.5, 2.6, 2.7: Mention and detail the use of positive and/or negative controls.

Results

Lines 154-158: This information is redundant with Figure 1. I suggest leaving only the text or the figure.

Figures 2 and 3: Same as the previous suggestion.

All tables and figures must also be mentioned in the text (i.e. Table 2, Figure 2).

Figure 6: Indicate the acronyms in detail in the footer.

An additional correlation analysis that could generate interesting information would be biofilm forming ability and virulence genes.

Discussion:

The discussions present notable deficiencies. The authors carried out different correlation analyzes between phenotypic and genotypic categories without adequate justification and support with references. Furthermore, most of the results on correlation are little or not discussed at all.

Lines 345-347: The authors cite an example that the presence of the gene is not a determinant of its activity, but rather its functionality. In this way, it is possible to notice that the results offer irrelevant information (i.e. Figure 5b) that does not allow us to differentiate between the phenotypes. This is a major limitation of the study and should be pointed out and corrected.

Others: 

Ethics committee statement: As a study includes sacrificed animals, approval by an ethics committee must be declared.

Mistyping: Check lines 82, 185, 205, 218.

The iThenticate report identified sentences and paragraphs with significant % identity with other manuscripts (i.e. https://www.mdpi.com/2076-2615/14/10/1433?) and should be corrected or appropriately cited.

Round 2

Reviewer 1 Report

Comments and Suggestions for Authors

All my comments have been addressed

Comments on the Quality of English Language

English has improved

Reviewer 3 Report

Comments and Suggestions for Authors

The authors have responded to or justified the observations noted.